# Species Identification of the Major Japanese Encephalitis Vectors within the *Culex vishnui* Subgroup (Diptera: Culicidae) in Thailand Using Geometric Morphometrics and DNA Barcoding

**DOI:** 10.3390/insects14020131

**Published:** 2023-01-26

**Authors:** Tawee Saiwichai, Sedthapong Laojun, Tanawat Chaiphongpachara, Suchada Sumruayphol

**Affiliations:** 1Department of Parasitology and Entomology, Faculty of Public Health, Mahidol University, Bangkok 10400, Thailand; 2Department of Public Health and Health Promotion, College of Allied Health Science, Suan Sunandha Rajabhat University, Samut Songkhram 75000, Thailand; 3Department of Medical Entomology, Faculty of Tropical Medicine, Mahidol University, Bangkok 10400, Thailand

**Keywords:** mosquito, *Culex vishnui* subgroup, geometric morphometrics, DNA barcoding, Thailand

## Abstract

**Simple Summary:**

*Culex pseudovishnui*, *Cx. tritaeniorhynchus*, and *Cx. vishnui* are members of the *Cx. vishnui* subgroup and are vectors of Japanese encephalitis (JE) in Thailand. However, their extremely similar morphologies make identification difficult. Thus, geometric morphometrics (GM) and DNA barcoding were applied to identify *Cx. pseudovishnui*, *Cx. tritaeniorhynchus*, and *Cx. vishnui* in Thailand. Our results of cross-validation reclassification demonstrated that GM based on wing shape was relatively high potential for distinguishing three *Culex* species. Meanwhile, DNA barcoding was highly effective at identifying all three species based on the DNA barcode gap.

**Abstract:**

Japanese encephalitis (JE) is a viral infection of the brain caused by the Japanese encephalitis virus, which spreads globally, particularly in 24 countries of Southeast Asia and the Western Pacific region. In Thailand, the primary vectors of JE are *Cx. pseudovishnui*, *Cx. tritaeniorhynchus*, and *Cx. vishnui* of the *Cx. vishnui* subgroup. The morphologies of three mosquito species are extremely similar, making identification challenging. Thus, geometric morphometrics (GM) and DNA barcoding were applied for species identification. The results of cross-validation reclassification revealed that the GM technique based on wing shape analysis had relatively high potential for distinguishing *Cx. pseudovishnui*, *Cx. tritaeniorhynchus*, and *Cx. vishnui* (total performance = 88.34% of correctly assigned individuals). While the DNA barcoding yielded excellent results in identifying these *Culex* species based on the DNA barcode gap (average intraspecific genetic distance = 0.78% ± 0.39% and average interspecific genetic distance = 6.14% ± 0.79%). However, in the absence of the required facilities for DNA barcoding, GM techniques can be employed in conjunction with morphological methods to enhance the reliability of species identification. Based on the results of this study, our approach can help guide efforts to identify members of the *Cx. vishnui* subgroup, which will be useful for the effective vector control of JE in Thailand.

## 1. Introduction

Japanese encephalitis (JE) is a brain infection caused by the Japanese encephalitis virus (JEV), which spreads globally, particularly in 24 Southeast Asian and Western Pacific countries [1]. JE has a 30% fatality rate, and its survivors often suffer from aphasia [2]. According to the World Health Organization (WHO), more than 3 billion people are at risk of JEV infection. In 2017, 2018, and 2019, the WHO recorded 4668, 4402, and 3709 global cases of JE, respectively [3]. Currently, there is no effective medication for JE, although at-risk populations can be protected by JE vaccination. Vaccine accessibility has become a significant public health issue, because numerous rural populations in many countries, including Thailand, still lack access to immunization [2]. According to data from the Bureau of Epidemiology, Thailand reported 14, 18, and 8 cases of JE in 2017, 2018, and 2019, respectively [4]. Patient morbidity of JE is not considered very high; however, the morbidity rates of mosquito-borne diseases in several Thai provinces are increasing [5]. Consequently, surveillance of JE is essential.

Mosquitoes pose a significant threat to global public health, particularly in tropical and subtropical regions, as they are the main vectors for transmitting deadly diseases to humans [6,7]. Most JE is spread by specific *Culex* species [8]. Of the 769 species of *Culex* mosquitoes worldwide [9], 82 are found in Thailand [10], where *Cx. vishnui* subgroup mosquitoes, comprising *Cx. pseudovishnui*, *Cx. tritaeniorhynchus*, and *Cx. vishnui*, are considered the principal JE vectors [10,11]. The habitats and breeding sites of these three *Culex* species comprise wells, marshy depressions, ditches, pits, sand pools, ground pools, stream pools, rice fields, footprints, wheel tracks, rock pools, and holes in natural containers [10]. In areas where mosquito-borne diseases are endemic, successful disease control requires knowledge of the vector species [12].

According to previous studies, species members of the *Cx. vishnui* subgroup exhibited distinct behaviors associated with disease outbreaks. For example, a survey of host selection or preference of mosquito species in Kandal province, Cambodia revealed that *Cx. vishnui* preferred humans more than animals, whereas *Cx. tritaeniorhynchus* preferred animals more than humans [13]. However, the three mosquito species that transmit JEV have very similar characteristics, making species identification problematic [10]. Normally, *Cx. tritaeniorhynchus* can be distinguished from *Cx. vishnui* and *Cx. pseudovishnui* based on a pale band on the proboscis. The proboscis of *Cx. tritaeniorhynchus* appears as a pale ring with a proximal extension on the ventral surface, which is absent in the *Cx. vishnui* and *Cx. pseudovishnui* proboscis [10]. Meanwhile, *Cx. pseudovishnui* can be distinguished from *Cx. vishnui* by a pale stripe on the anterior surface of the hindfemur of *Cx. vishnui*, which does not distinctly contrast from the dark-scaled area, whereas this pale stripe contrasts well with the dark-scaled area in *Cx. pseudovishnui* [10]. These important characteristics of the three *Culex* species are fragile and easily damaged, especially during the act of collecting the adult mosquitoes in the field using mosquito traps, making it challenging to identify species morphologically [14].

Successfully controlling these three *Culex* populations can be difficult due to their incorrect identification. There are several effective methods for controlling mosquito vectors; however, selection of the appropriate method relies on an accurate identification of the mosquito species, because most methods rely on species-specific characteristics and behaviors, such as larval habitat and feeding and resting behaviors [12,15]. The standard method for species identification of the *Cx. vishnui* subgroup relies on morphological traits that are unique to each species. However, this method is limited by the need for complete morphological specimens and practitioner expertise [10]. Currently, effective alternative methods for species identification include geometric morphometrics and DNA barcoding; both may support the standard method to accurately identify mosquito samples that are morphologically similar [16,17,18,19,20,21,22].

Geometric morphometrics (GM) is a modern alternative technique for identifying mosquito species. It is gaining popularity because it is quick, inexpensive, and does not require advanced laboratory equipment [23]. GM is based on the analysis and comparison of structural variations, including the shape and size of organisms [23,24,25]. The landmark-based GM approach is the most popular, and it is based on the analysis of the size and shape of wing based on the positions of anatomical landmark coordinates (also called true landmarks) and the distance between landmarks that contains both size and shape data [23,24]. The results of previous research carried out in Thailand demonstrated that the GM technique is an effective tool for identifying various mosquito species [15,17,20,26,27,28].

DNA barcoding refers to one of the molecular techniques that uses a short section of DNA from a specific gene in eukaryotic organisms for rapid species identification, especially among members of the kingdom Animalia [29]. The mitochondrial cytochrome *c* oxidase subunit I (*COI*) gene is frequently used as the marker of choice for DNA barcoding [30] and metabarcoding approach [31]. Currently, DNA barcodes are being widely used to identify several animal species, particularly insects in the order Diptera. Moreover, this technique has been very successful in identifying mosquitoes in Thailand [21,28].

Here, we applied DNA barcoding and the landmark-based GM approach to identify *Cx. pseudovishnui*, *Cx. tritaeniorhynchus*, and *Cx. vishnui*. Based on the results of this study, our approach can help guide efforts to identify members of the *Cx. vishnui* subgroup, which will be useful for the effective vector control of JE in Thailand.

## 2. Materials and Methods

### 2.1. Mosquito Collections

Adult *Cx. pseudovishnui*, *Cx. tritaeniorhynchus*, and *Cx. vishnui* were collected from three provinces in Thailand: Nakhon Pathom (13°53′29.6″ N, 100°00′08.8″ E), Kanchanaburi (14°07′06.2″ N, 99°01′16.2″ E), and Ratchaburi (13°21′35.1″ N, 99°15′04.5″ E) (Figure 1). Ten BG-Pro CDC-style traps (Biogents, Regensburg, Germany) used in with dry ice and BG-lure (Biogents, Regensberg, Germany) were used in trapping adult mosquito for ten consecutive nights per site during July and September 2021 between 6 p.m. (start time) and 6 a.m. (end time). Ten traps were hung at heights of 1.5 and 5 m around cottages near rice fields, which are the breeding sites for mosquitoes in the *Cx. vishnui* subgroup. In the morning (6 a.m. (end time)), the field collecting bag was taken out of the trap and cooled for approximately 30 min in an ice box containing 1 kg of dry ice to kill all mosquitoes. 

This entomological study was approved by the Institute of Animal Care and Use Committee (MU-IACUC) of the Faculty of Tropical Medicine, Mahidol University, Thailand (reference no. 008-2021E).

### 2.2. Species Identification Based on the Morphology

Mosquitoes were transported from the field to the laboratory of the College of Allied Health Sciences, Suan Sunandha Rajabhat University, Samut Songkhram Campus for species identification based on morphological characters. All samples were stored at −20 °C in the laboratory’s freezer pending species identification. Adult female *Culex* mosquitoes were examined under a stereomicroscope (Nikon SMZ 745; Nikon Corp., Tokyo, Japan) and identified using illustrated keys to the medically important mosquitoes of Thailand [10]. Only complete *Culex* specimens without damaged unique portions and morphologically identifiable were used for analyses. While morphologically ambiguous specimens were excluded from this experiment.

### 2.3. Sample Preparation for GM and DNA Barcoding

The right wings of *Cx. pseudovishnui*, *Cx. tritaeniorhynchus*, and *Cx. vishnui* were used for the GM analysis, whereas their legs were used for DNA barcoding. The wings of mosquitoes were dissected from the thorax, placed on a microscope slide, and then mounted in Hoyer’s mounting medium solution. The plume scales were removed using bristle needles, and the specimen was covered with a cover slip. Images of all wing slides were captured using a (Nikon SMZ 800N) digital camera coupled to a stereomicroscope at 40× magnification using the NIS-Elements application. Additionally, a 1 mm scale bar was placed in the upper left corner of each wing image.

The legs of 10 mosquitoes per species were randomly collected from samples that were used for the GM analysis. The legs were placed in a 1.5 mL tube (one individual per one tube) pending DNA extraction.

### 2.4. Geometric Morphometrics

In this study, we used XYOM (freely available at https://xyom.io/ (accessed on 4 March 2022)) for GM analysis [32]. Eighteen anatomical landmarks on the wing veins were digitized (Figure 2) according to criteria used in previous studies [17,18,33].

After that, epeatability was computed to ensure the precision of landmark digitization. Two sets of wing images were used to find the value of shape repeatability using Procrustes analysis [34]. Then, the relationship between the wing size and wing shape of all samples was determined by allometric examination, which involves the calculation of the linear determination coefficient after regressing principal components of wing shape on wing size.

The wing size is expressed as the centroid size (CS), which is the result of the square root of the sum of the squared distances from the centroid to the 18 wing landmarks [35]. The mean wing sizes among the three *Culex* species were evaluated by a nonparametric ANOVA with 1000 permutations. Statistical significance was set at *p* < 0.05.

The variables for wing shape were calculated by principal component analysis of the partial warp after the generalized Procrustes analysis [36,37]. After that, discriminant analysis and Mahalanobis distance were computed using the final shape variables. The Mahalanobis distance refers to the statistical distance between the wing shapes of the three *Culex* species. Wing shape difference based on Mahalanobis distances between *Culex* species were compared using a nonparametric test (1000 permutations) with Bonferroni correction at *p*-values < 0.05.

The accuracy of species identification based on wing size was validated by maximum likelihood-based classification [38], whereas the accuracy of identification based on wing shape was validated by Mahalanobis-based classification [39]. For cross-validation reclassification, each individual is iteratively excluded from the total sample and assigned to the closest group to see how resampling the dataset affects the assignment estimation. A hierarchical clustering tree based on the Mahalanobis distances was constructed to examine the relationship between *Culex* species based on wing shape.

### 2.5. DNA Barcoding

Thirty *Culex* samples composed of 10 individuals each of *Cx. pseudovishnui*, *Cx. tritaeniorhynchus*, and *Cx. vishnu* were used for DNA barcoding analysis. The DNA from at least 2 legs per individual of all 30 samples were extracted using the FavorPrep™ tissue genomic DNA extraction mini kit (Favorgen Biotech, Ping-Tung, Taiwan) following the manufacturer’s instructions. PCR was used to amplify a fragment of the *COI* gene using forward (5′GGATTTGGAAATTGATTAGTTCCTT3′) and reverse (5′AAAAATTTTAA TTCCAGTTGGAACAGC3′) barcoding primers [40]. Each PCR reaction consisted of a 25-µL solution containing 4 µL of DNA, 1.5 unit of Platinum Taq DNA polymerase (Invitrogen), 0.2 μM of each primer, 0.2 mM of dNTPs, and 1x reaction buffer, 1.5 mM MgCl_2_, 5% dimethyl sulfoxide, and distilled water up to 25 μL. The PCR cycles consisted of 95 °C for 5 min, followed by 5 cycles at 94 °C for 40 s, 45 °C for 60 s, and 72 °C; 35 cycles at 94 °C for 40 s, 54 °C for 60 s, and 72 °C for 1 min; and a final extension at 72 °C for 10 min.

The PCR products were visualized by electrophoresis in 1.5% agarose gels with Midori Green Advance DNA staining solution (Nippon Gene, Tokyo, Japan). The Gel Doc™ image Quant LAS 500 (GE Healthcare Bio-Sciences AB, Uppsala, Sweden) was used to examine and photograph DNA bands on agarose gels to evaluate the quality of PCR products. PCR products of sufficient quality were then submitted to SolGent Co., Ltd. (Daejeon, Republic of Korea) for purification and sequencing.

After Sanger sequencing, chromatogram trace files of *COI* sequences were manually edited, then ambiguous sites were removed, and consensus sequences were created by BioEdit software. To confirm the identities our *COI* sequences, they were compared to sequences published in GenBank (https://blast.ncbi.nlm.nih.gov/, accessed on 11 April 2022) and BOLD Systems (https://www.boldsystems.org/index.php/, accessed on 11 April 2022). DNA sequences were aligned using the Clustal W tool [41] within Molecular Evolutionary Genetics Analysis (MEGA) version X [42]. Intra- and interspecific pairwise distances based on the Kimura two-parameter (K2P) model for the *COI* barcodes of *Cx. pseudovishnui*, *Cx. tritaeniorhynchus*, and *Cx. vishnui* were calculated using MEGA X. Furthermore, the nucleotide sequences were used to build phylogenetic trees based on the neighbor-joining (NJ) method with bootstrapping (1000 replications) by MEGA X to determine the phylogenetic relationships among *Cx. pseudovishnui*, *Cx. tritaeniorhynchus*, and *Cx. vishnui*. *Culex quinquefasciatus* (OL743030) was included in the phylogenetic tree as an outgroup. The FigTree v.1.4.3 (available at http://tree.bio.ed.ac.uk/software/Figtree/, accessed on 2 May 2022) was used to decorate, modify and create a phylogenetic tree.

For molecular species delimitation, the online assemble species by automatic partitioning (ASAP) approach based on the simple distance (*p*-distances) was used (https://bioinfo.mnhn.fr/abi/public/asap/, accessed on 10 May 2022). Our *COI* sequences were submitted to the GenBank database under accession numbers OK147859–OK147868 (for *Cx. pseudovishnui*), OK147839–OK147848 (for *Cx. tritaeniorhynchus*), and OK147849–OK147858 (for *Cx. vishnui*).

## 3. Results

### 3.1. Geometric Morphometrics for Species Identification

A total of 163 wings of the *Cx. vishnui* subgroup were analyzed by GM, including 40, 58, and 65 wings of *Cx. pseudovishnui*, *Cx. tritaeniorhynchus*, and *Cx. vishnui*, respectively. The repeatability value of the wing image set was 95%, while the measurement error was 5%, indicating that the quality of our landmark digitization was high. 

The relationship between size and shape (allometry) was estimated using the scatter diagram in Figure 3, which indicates a linear relationship between the two variables. The slope of the scatter diagram indicates a negative linear relationship between size and shape, with an r^2^ value of 11.60 (*p* < 0.05).

Analysis of wing size variation based on wing CS calculations of *Cx. pseudovishnui*, *Cx. tritaeniorhynchus*, and *Cx. vishnui* is shown in Figure 4. According to wing CS analysis, *Cx. vishnui* had the largest wings (CS = 2.58 mm), followed by *Cx. pseudovishnui* (CS = 2.45 mm) and *Cx. tritaeniorhynchus* (CS = 2.23 mm). Results of statistical analysis confirmed the differences in wing size of the three *Culex* species. (*p* < 0.05, Table 1).

Establishing the superposition of the landmark sites on the wings of the *Cx. vishnui* subgroup after alignment, as depicted in Figure 5, reveals the various different characteristics of wing shape between *Culex* species. 

Figure 6 shows a factor map based on discriminant analysis of shape variables, and it shows that wing shape varies between *Cx. pseudovishnui*, *Cx. tritaeniorhynchus*, and *Cx. vishnui*, i.e., although the group space of three *Culex* species have overlapping distributions, the wing shapes differ significantly based on the pairwise Mahalanobis distances (1000 runs, *p* < 0.05, Table 2). The hierarchical clustering tree based on the wing shapes of the three *Cx. vishnui* subgroup species is shown in Figure 7.

Analysis of wing size based on centroid size reclassification of the *Cx. vishnui* subgroup shows a lower total percentage score (58.28%) compared to the cross-validated reclassification based on wing shape obtained from Mahalanobis distance values (88.34%, Table 3). 

According to the validated reclassification based on wing shape, *Cx. tritaeniorhynchus* had the highest percentage of correctly classified individual (98.28%), followed by *Cx. vishnui* (87.69%), and *Cx. pseudovishnui* (75%).

### 3.2. Species Identification by DNA Barcoding

DNA barcoding was used in parallel with the GM approach for the identification of *Cx. pseudovishnui*, *Cx. tritaeniorhynchus*, and *Cx. vishnui*. The length of the mitochondrial *COI* sequences analyzed was 735 bp. The average ATGC composition of 30 *Culex* sequences were 30.3%, 37.7%, 16.1%, and 15.8%, respectively. Sequence comparisons to the GenBank and Bold databases indicated that the identities of our mosquito species based on morphology were correct (>98% sequence similarity).

The average intraspecific genetic distance among the *Culex* species based on K2P distance calculations was 0.78% ± 0.39% (average ± SD). The highest average intraspecific divergence was found 0.90% ± 0.40% in *Cx. triteaniorhynchus*, followed by *Cx. pseudovishnui* at 0.81% ± 0.31% and *Cx. vishnui* at 0.67% ± 0.41% (Table 4). Meanwhile, the average interspecific genetic distance was 6.14% ± 0.79%. The highest average interspecific divergence was between *Cx. triteaniorhynchus* and *Cx. pseudovishnui* (6.87% ± 0.30%), followed by *Cx. triteaniorhynchus* and *Cx. vishnui* (5.40% ± 0.27%) and *Cx. vishnui* and *Cx. pseudovishnui* (4.71% ± 0.43%). 

An NJ tree based on *COI* sequences revealed that *Cx. vishnui* and *Cx. pseudovishnui* formed a well-supported clade that was sister to *Cx. tritaeniorhynchus* (bootstrap support > 93%, Figure 8). The ASAP procedure indicated concordant results with morphological taxonomy (the “best” ASAP score = 1.00 and a threshold distance = 0.025).

## 4. Discussion

The use of morphology to identify mosquito species is the most commonly used method of identification. However, this method is prone to error, especially when identifying mosquitoes belonging to a species complex, as well as isomorphic species, cryptic species, and samples damaged during collection [12]. Therefore, several more advanced and accurate modern techniques have been used to identify mosquitoes, and these have greatly increased our knowledge and understanding of mosquito vectors [16,17,18,19,20,21,22]. *Culex pseudovishnui*, *Cx. tritaeniorhynchus*, and *Cx. vishnui* are the major vectors of JE and are common in many Asian countries [43,44]. However, the morphology of each of these *Culex* species is slightly different, and specimens tend to be easily damaged [10]. This study is the first to use the landmark-based GM approach to distinguish between *Cx. pseudovishnui*, *Cx. tritaeniorhynchus*, and *Cx. vishnui*. Furthermore, we applied DNA barcoding in parallel with the GM approach to identify these three *Culex* species.

Only undamaged samples are used in the GM analysis to reduce sources of error. The result of the allometric examination based on the linear determination coefficient shows that wing size and wing shape of our *Culex* samples are negatively correlated. Similar observations have been reported for several mosquito species, such as *Anopheles dissidens*, *An. wejchoochotei*, and *An. saeungae* in Thailand [18]. Although the relationship between wing size and shape impacts sexual shape dimorphism, it has no effect on species identification based on landmark-based GM analysis [23].

The results of GM analysis showed that the wings of all three *Culex* species differ significantly in size. Nevertheless, the maximum likelihood-based validated classification based on wing size yielded poor results in distinguishing between the three *Culex* species (total performance = 58.28%). This result is consistent with the findings of previous research, indicating that wing size is unsuitable for the identification of mosquito species [17,18,27,28]. The wing size is more sensitive to the environment, particularly the influence of temperature, humidity, and food availability in breeding sites, during the development of the immature stage of mosquito [28]. In contrast, wing shape analyses yielded better outcomes than wing size.

The validated classification based on wing shape (total performance = 88.34% of correctly assigned individuals) clearly outperformed wing size analyses. This is consistent with previous studies that applied the GM approach in identifying mosquito vectors and found that wing shape is more effective criterion compared to wing size [27,28]. Several studies have demonstrated that the GM technique is effective for the identification of mosquito species in Thailand, including *Aedes* species (*Ae. aegypti*, *Ae. albopictus*, and *Ae. scutellaris* [27,45]); *Mansonia* species (*Ma. dives* and *Ma. bonneae* [28]); *Culex* species (*Cx. visnui*, *Cx. sitiens*, and *Cx. whitmorei* [14]); and *Anopheles* species (*An. dirus*, *An. baimaii* [17], *An. maculatus*, *An. sawadwongporni*, and *An. pseudowillmori* [18]). Nevertheless, a performance level of 88.34%, as we found, is not considered perfect but still has a relatively high potential. The results of discriminant analysis indicate that *Cx. pseudovishnui*, *Cx. tritaeniorhynchus*, and *Cx. vishnui* share very similar wing characteristics that overlap among species. If species identification is limited by financial constraints, then the GM technique based on wing shape can be used in conjunction with the morphological method, which is better than visual inspection alone [28].

DNA barcoding is a highly effective molecular technique for the identification of mosquito species that are morphologically similar or indistinguishable [21,22,46,47]. Our genetic analysis and the ASAP method suggested that DNA barcoding based on *COI* sequences can be used to distinguish between *Cx. pseudovishnui*, *Cx. tritaeniorhynchus*, and *Cx. vishnui* and the results were consistent with those of morphological identification. Analysis of the genetic distance between species shows that the species are separated by a DNA barcoding gap, which is the difference between the greatest intraspecific distance and the smallest interspecific distance [21,22,46]. Our results of DNA barcoding are consistent with those of a previous study in Sri Lanka that used DNA barcoding to successfully identify members in the *Cx. vishnui* subgroup [30]. In addition, the NJ tree based on *COI* sequences is consistent with the hierarchical clustering tree based on wing shape (GM) analysis. Both trees indicate that the *Cx. tritaeniorhynchus* clade is distinct from the other two clades, while *Cx. vishnui* and *Cx. pseudovishnui* are more closely related to each other. A previous study that applied GM and genetic techniques to compare variability between male and female *Ae. aegypti* populations in the Philippines found that phenotypic and genetic patterns in male *Ae. aegypti* are highly congruent [48], suggesting that wing shape is genetically influenced.

## 5. Conclusions

The cross-validation reclassification revealed that the GM technique had relatively high potential for distinguishing *Cx. pseudovishnui, Cx. tritaeniorhynchus*, and *Cx. vishnui* based on wing shape analysis (88.34% of correctly assigned individuals). While the DNA barcoding yielded excellent results in identifying these *Culex* species based on the DNA barcode gap. Therefore, we suggest that DNA barcoding based on the *COI* gene is an excellent method for identifying *Cx. pseudovishnui*, *Cx. tritaeniorhynchus*, and *Cx. vishnui*, which are difficult to identify morphologically. However, in the absence of the required facilities for DNA barcoding, GM techniques can be employed in conjunction with morphological methods to enhance the reliability of species identification. The GM technique offers an inexpensive, and rapid analysis of species identification (approximately 5–10 min). Currently, there are several software freely available for studying geometric morphometrics. In addition, it can be implemented using basic equipment found in most entomology laboratories. However, the wing slide preparation is a rather time-consuming process (approximately 10 wings per hour). Our results will help guide efforts to identify members of the *Cx. vishnui* subgroup, which is a requirement for the effective control of the vector of JE in Thailand.

## Figures and Tables

**Figure 1 insects-14-00131-f001:**
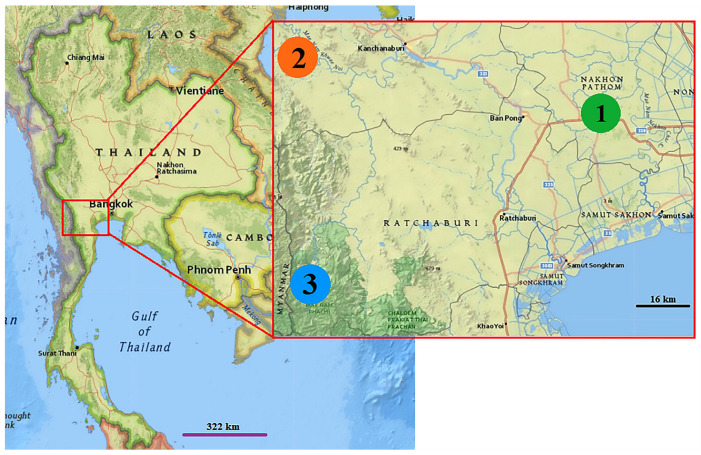
Study sites for mosquito sample collection in this study. This geographical map was taken from the USGS National Map Viewer (the public domain) at http://viewer.nationalmap.gov/viewer/ (accessed on 9 November 2022). Mosquito samples were collected from Nakhon Pathom (1, Green) in the central region, Kanchanaburi (2, Orange), and Ratchaburi (3, Blue) in the western region of Thailand.

**Figure 2 insects-14-00131-f002:**
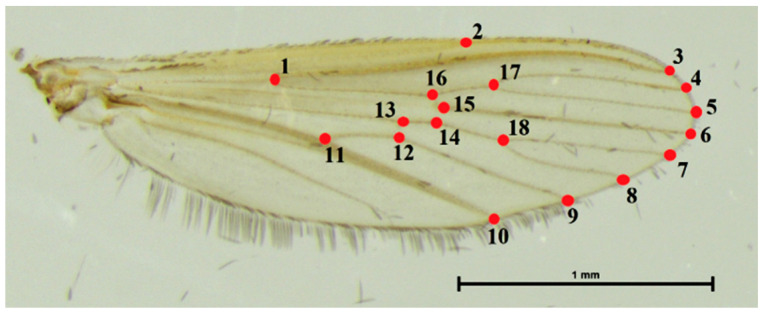
Location of the 18 wing landmarks used in the geometric morphometric analyses.

**Figure 3 insects-14-00131-f003:**
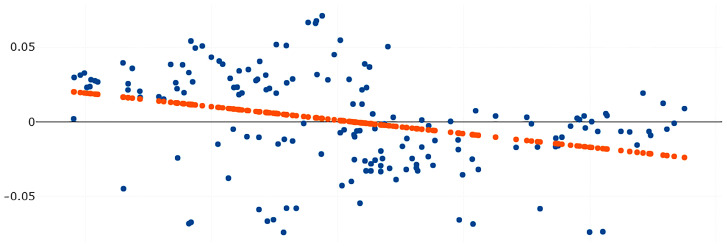
Scatter diagram showing the relationship between wing size (the *x*-axis) and wing shape (the *y*-axis) using landmark-based GM analysis. The linear regression prediction result is represented by the orange dots. While blue dots indicate individual mosquitoes.

**Figure 4 insects-14-00131-f004:**
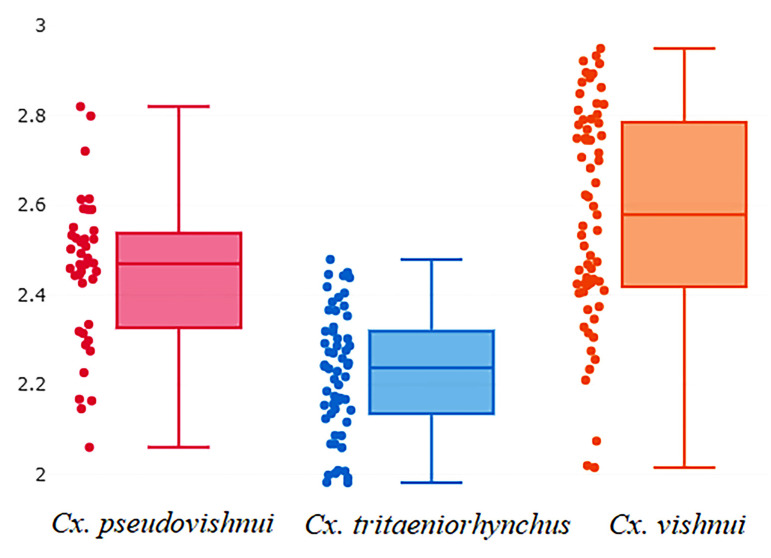
Boxplot of the wing centroid size of *Cx. pseudovishnui*, *Cx. tritaeniorhynchus*, and *Cx. vishnui*. Each box represents the wing size data of each *Culex* species, and the group median that separates the 25th and 75th quartiles is displayed in each box.

**Figure 5 insects-14-00131-f005:**
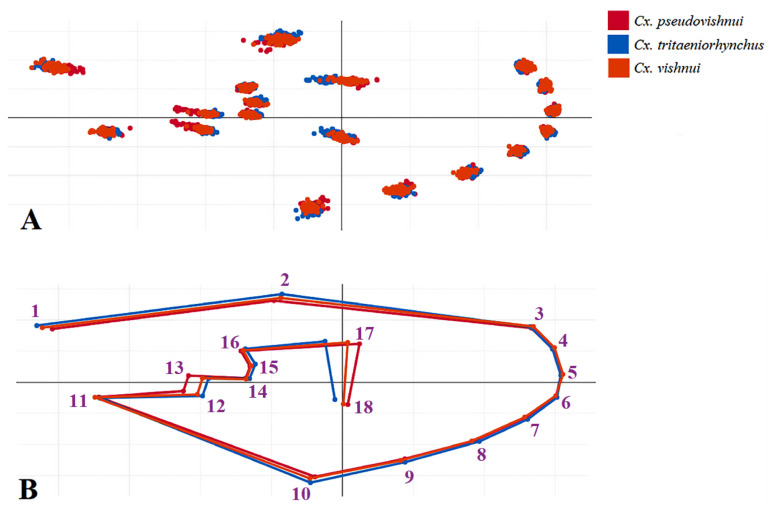
Rotated landmarks of aligned objects (**A**) and the mean landmark superposition (**B**) of *Cx. pseudovishnui*, *Cx. tritaeniorhynchus*, and *Cx. vishnui* wings.

**Figure 6 insects-14-00131-f006:**
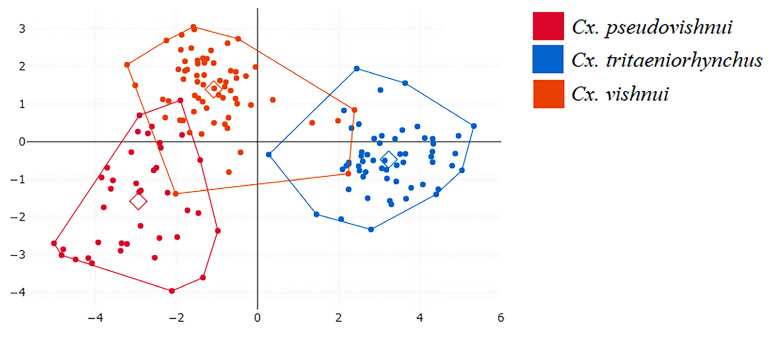
Factor map generated by discriminant analysis of shape variables for *Cx. pseudovishnui*, *Cx. tritaeniorhynchus*, and *Cx. vishnui*. The wing shape variations of each *Culex* species are represented by each polygon, and each dot within the polygons represents an individual *Culex* specimen.

**Figure 7 insects-14-00131-f007:**
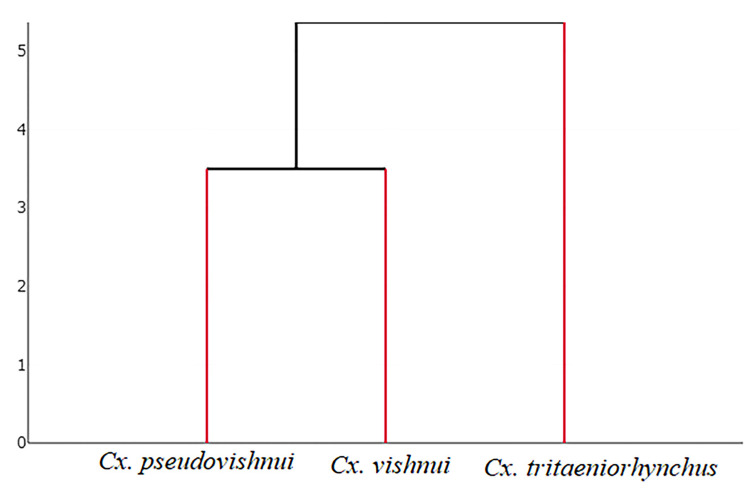
Hierarchical clustering tree based on wing shape among *Cx. pseudovishnui*, *Cx. tritaeniorhynchus*, and *Cx. vishnui*. The UPGMA algorithm is used to create a hierarchical clustering tree based on Mahalanobis distances between average group shapes.

**Figure 8 insects-14-00131-f008:**
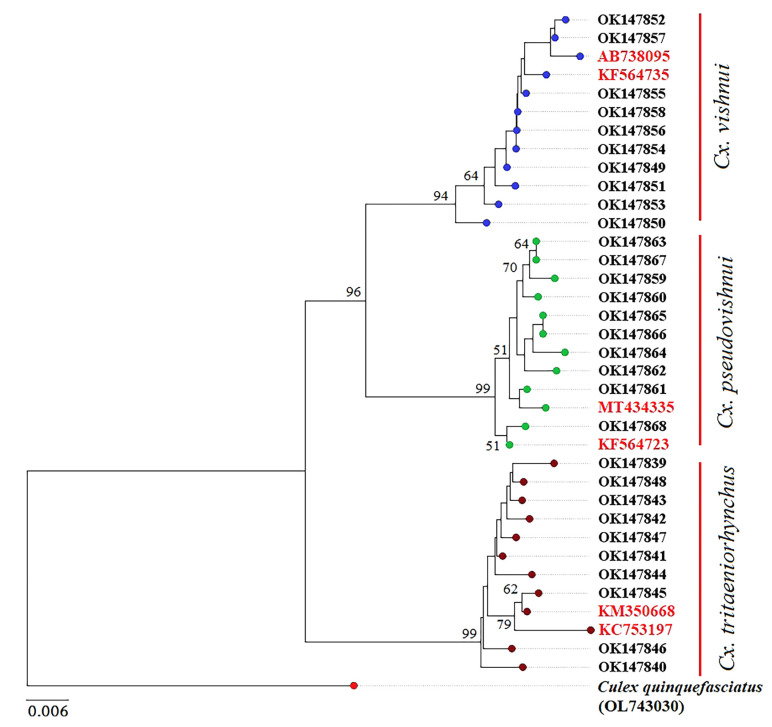
Neighbor-joining tree showing the phylogenetic relationships between *Cx. pseudovishnui*, *Cx. tritaeniorhynchus*, and *Cx. vishnui*. The *COI* sequences generated in this study are black (n = 30), while those from GenBank are red (n = 6). The *Cx. quinquefasciatus* sequence from GenBank (OL743030) was used as an outgroup. Only bootstrap values ≥ 50% are shown. The result of species delimitation by the ASAP method is indicated by vertical red bars.

**Table 1 insects-14-00131-t001:** Statistical differences in mean wing centroid size of *Cx. pseudovishnui*, *Cx. tritaeniorhynchus*, and *Cx. vishnui*.

Species	*n*	Mean (mm)	Min–Max	SD	SE
*Cx. pseudovishnui*	40	2.45 ^a^	2.06–2.82	0.17	0.03
*Cx. tritaeniorhynchus*	58	2.23 ^b^	1.98–2.48	0.14	0.02
*Cx. vishnui*	65	2.58 ^c^	2.02–2.95	0.24	0.03

Different superscript letters after wing CS values indicate significant differences at *p* < 0.05.

**Table 2 insects-14-00131-t002:** Statistical differences of Mahalanobis distance values based on wing shapes of *Cx. Pseudovishnui*, *Cx. tritaeniorhynchus*, and *Cx. vishnui*.

Species	*Cx. pseudovishnui*	*Cx. tritaeniorhynchus*	*Cx. vishnui*
*Cx. pseudovishnui*	0.000		
*Cx. tritaeniorhynchus*	6.268 *	0.000	
*Cx. vishnui*	3.498 *	4.695 *	0.000

The superscript asterisks after Mahalanobis distance values indicate significant differences between *Culex* species at *p* < 0.05.

**Table 3 insects-14-00131-t003:** Cross-validation reclassification score (%) based on wing size (centroid size) and wing shape (Mahalanobis distance values) of the *Cx. vishnui* subgroup.

Species	Cross-Validated Reclassification Score (%)	
	Wing Size (Assigned/Observed)	Wing Shape (Assigned/Observed)
*Cx. pseudovishnui*	35.00% (14/40)	75.00% (30/40)
*Cx. tritaeniorhynchus*	75.86% (44/58)	98.28% (57/58)
*Cx. vishnui*	56.92% (37/65)	87.69% (57/65)
Total performance	58.28% (95/163)	88.34% (144/163)

**Table 4 insects-14-00131-t004:** Inter- and intraspecific genetic divergence between *Cx. pseudovishnui*, *Cx. tritaeniorhynchus*, and *Cx. vishnui* based on the Kimura two parameter model.

Species	Average Percentage Genetic Divergence (Min–Max)		
	*Cx. pseudovishnui*	*Cx. tritaeniorhynchus*	*Cx. vishnui*
*Cx. pseudovishnui*	0.81% (0.00–1.43)		
*Cx. tritaeniorhynchus*	6.87% (6.33–7.89)	0.90% (0.28–1.86)	
*Cx. vishnui*	4.71% (3.91–5.90)	5.40% (4.96–6.03)	0.67% (0.00–1.57)

## Data Availability

Our *COI* sequences were submitted in the GenBank database with the following accession numbers: *Cx. pseudovishnui*: OK147859–OK147868; *Cx. tritaeniorhynchus*: OK147839–OK147848; *Cx. vishnui*: OK147849–OK147858.

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
