# Peer review of "Species Identification of the Major Japanese Encephalitis Vectors within the Culex vishnui Subgroup (Diptera: Culicidae) in Thailand Using Geometric Morphometrics and DNA Barcoding"

_insects, 2023, doi:10.3390/insects14020131_

Round 1

Reviewer 1 Report

This manuscript applies geometric morphometrics (GM) and DNA barcoding to address the difficulties often experienced in separating Cx. tritaeniorynchus, Cx. vishnui and Cx. pseudovishnui using standard morphological techniques.  The authors are correct in stating that field-collected adult mosquitoes of these species often have key morphological characteristics missing or damaged.  Previous publications have addressed this using molecular techniques (e.g., the PCR protocols described in reference 15), and DNA barcoding has become widely adopted to identify morphologically similar specimens.  To my knowledge, this is the first time GM has been applied to the Cx. vishnui subgroup.  The results of this project demonstrate the high level of accuracy barcoding can achieve and demonstrate that the GM procedure they used, though considerably less accurate, may help increase identification accuracy in resource-limited settings where molecular techniques are not feasible.  The experimental design and analyses are appropriate, and the conclusions are generally supported by the results.  I think the manuscript would benefit from more information regarding the steps that would be required to modify this to an operational JE surveillance program.   

 I have a few comments the authors should consider:

Materials and Methods:

Line 131 states that traps were hung at heights of 1.5 and 50 meters.  I assume the latter should be 5.0 meters.

 Lines 143-146 note that adult female mosquitoes were identified using morphological characteristics and illustrated keys.  Given the previously described difficulties in separating these species using morphological techniques, it is not clear why this method was used to obtain accurately identified specimens subsequently used in the GM and barcoding analyses.  However, results (lines 329-331) indicate that the initial morphological identification was >98% correct.  This seems to contradict the statements about lack of accuracy using morphological characters.  I suspect that the specimens identified and selected for the subsequent analyses were pristine and the taxonomist performing the identifications had a high level of confidence in the species determination.  I think it would be useful to give the reader an idea of how many specimens were screened to yield the 163 selected for the subsequent analyses.

 Discussion:

Lines 359-360 refers to morphological identification as the “gold standard” method of identification.  I will agree that it is the most commonly used method of identification, but it is not the gold standard, which implies that it is the best or most reliable method.  Rather, molecular techniques like DNA barcoding have become the gold standard due to their accuracy and sensitivity.

Conclusions:

Lines 440-442 suggest that GM is quick, inexpensive, rapid, and can be conducted using basic equipment found in most labs.  I think readers would benefit from an expanded description of the time required to mount and photograph the wings, and to digitize the measurements and run the analyses.  Also, it is unclear if the operational users must conduct their own statistical analyses, or if standards for species identification can be derived from this manuscript. 

Author Response

Response to reviewers

Reviewer #1:

Comments and Suggestions for Authors

This manuscript applies geometric morphometrics (GM) and DNA barcoding to address the difficulties often experienced in separating Cx. tritaeniorynchus, Cx. vishnui and Cx. pseudovishnui using standard morphological techniques.  The authors are correct in stating that field-collected adult mosquitoes of these species often have key morphological characteristics missing or damaged.  Previous publications have addressed this using molecular techniques (e.g., the PCR protocols described in reference 15), and DNA barcoding has become widely adopted to identify morphologically similar specimens.  To my knowledge, this is the first time GM has been applied to the Cx. vishnui subgroup. The results of this project demonstrate the high level of accuracy barcoding can achieve and demonstrate that the GM procedure they used, though considerably less accurate, may help increase identification accuracy in resource-limited settings where molecular techniques are not feasible. The experimental design and analyses are appropriate, and the conclusions are generally supported by the results. I think the manuscript would benefit from more information regarding the steps that would be required to modify this to an operational JE surveillance program.  

  1. Line 131 states that traps were hung at heights of 1.5 and 50 meters. I assume the latter should be 5.0 meters.

- Thank you very much. I fixed it as you suggested. “Ten traps were hung at heights of 1.5 and 5 meters around cottages near rice fields, which are the breeding sites for mosquitoes in the Cx. vishnui subgroup.”

  1. Lines 143-146 note that adult female mosquitoes were identified using morphological characteristics and illustrated keys. Given the previously described difficulties in separating these species using morphological techniques, it is not clear why this method was used to obtain accurately identified specimens subsequently used in the GM and barcoding analyses. However, results (lines 329-331) indicate that the initial morphological identification was >98% correct.  This seems to contradict the statements about lack of accuracy using morphological characters.  I suspect that the specimens identified and selected for the subsequent analyses were pristine and the taxonomist performing the identifications had a high level of confidence in the species determination.  I think it would be useful to give the reader an idea of how many specimens were screened to yield the 163 selected for the subsequent analyses.

- Many thanks, I added as you suggested in the “Species identification based on the morphology” section. “Only complete Culex specimens without damaged unique portions and morphologically identifiable were used for analyses. Morphologically ambiguous specimens were excluded from this experiment.”

  1. Lines 359-360 refers to morphological identification as the “gold standard” method of identification.  I will agree that it is the most commonly used method of identification, but it is not the gold standard, which implies that it is the best or most reliable method.  Rather, molecular techniques like DNA barcoding have become the gold standard due to their accuracy and sensitivity.

- Thank you very much. I fixed it as you suggested. “The use of morphology to identify mosquito species is the most commonly used method of identification.”

  1. Lines 440-442 suggest that GM is quick, inexpensive, rapid, and can be conducted using basic equipment found in most labs.  I think readers would benefit from an expanded description of the time required to mount and photograph the wings, and to digitize the measurements and run the analyses.  Also, it is unclear if the operational users must conduct their own statistical analyses, or if standards for species identification can be derived from this manuscript. 

- Many thanks, I added as you suggested in the “Conclusions” section. “The GM technique offers an inexpensive, and rapid analysis of species identification (approximately 5-10 minutes). Currently, there are several software freely available for studying geometric morphometrics. In addition, it can be implemented using basic equipment found in most entomology laboratories. However, the wing slide preparation is a rather time-consuming process (approximately 10 wings per hour). Our results will help guide efforts to identify members of the Cx. vishnui subgroup, which is a requirement for the effective control of the vector of JE in Thailand.”

Reviewer 2 Report

The MS concerns distinguishing between three mosquito species - Culex pseudovishnui, Cx. tritaeniorhynchus, and Cx. vishnui - that belong to the Cx. vishnui species complex. However, the authors do not argue the need to distinguish between these species from an epidemiological point of view, as all three species are closely-related and all can spread Japanese encephalitis virus. Maybe they can add in the introduction what is the epidemic benefit of distinguishing between these species instead of identifying Cx. vishnui sensu lato?

 Overall, the study is done correctly, with the main result being that geometric morphometric (GM) analysis does not provide a guarantee of species differentiation in the complex. While I cannot evaluate morphological analysis, in my opinion a formal barcoding-gap analysis should be used for DNA-barcode analysis (https://bioinfo.mnhn.fr/abi/public/asap/, https://onlinelibrary.wiley.com/doi/10.1111/1755-0998.13281).

I also have some minor comments that should be taken into account before accepting this manuscript:

1. Throughout the manuscript, the three species should be given in alphabetical order (Culex pseudovishnui, Cx. tritaeniorhynchus, and Cx. vishnui); the information that Cx. pseudovishnui and Cx. vishnui are more closely related is only the result of this study.

2. L15, 64 – Culex should be abbreviated.

3. The abstract (L36-37) is not consistent with the results and the conclusion that GM does not guarantee species differentiation.

4. L112, at the end of the sentence that COI is frequently used as the marker of choice for DNA barcoding you can add “including metabarcoding approach” (you can see and cite Trzebny et al. 2020, https://doi.org/10.1111/1755-0998.13205).

5. L216-219 - It is better to remove gaps and ends (5’ and 3’) from the primer sequences, as they make sequences harder to copy for ordering primers.

6. Line 218 – remove the word “mix”.

7. L220 - Instead "0.2 μM forward primer, 0.2 μM reverse primer" just "0.2 μM each primer".

8. L230 - remouve "nucleotide" just "sequencing".

9. L235-237 - change to: "were compared to sequences published in GenBank (https://blast.ncbi.nlm.nih.gov/) and BOLD Systems (https://www.boldsystems.org/index.php)."

10. L248 – some fonts are bigger than others. 

11. The caption to Figure 7 should be expanded to include information on what data was used and how they were analyzed.

12. L327 – sequences cannot be “approximately 735 bp” long. They have to be the same length (it is impossible that deletion/insertion of an amino acid can occur in such closely related species); they also should be trimmed to the same length in order not to introduce additional, non-existent sequence variability.

13. L332-345 – Distances should include standard deviation (SD) value instead of a range. Also phylogenetic relationships could be better described (e.g. Cx. vishnui and Cx. pseudovishnui formed well supported clade which was sister to Cx. tritaeniorhynchus).

14. Table 4 repeats content from L331-341.

15. Fig. 8 caption – remove “text of the” and leave only “The COI sequences...”. Why “Culex species” is underlined?

16. L390 – check the sentence; something is missing (“immature stage of 28].”). The next sentence is also unclear, and information is tautological.

17. L432 - and everywhere in the text where it fits - maybe it's a good idea to replace species identification with species distinguishing?

18. The conclusions should be re-written. They should not provide results and should include a real finding that the GM is suitable for identifying Cx. vishnui sensu lato, but particular species in the complex cannot be unambiguously distinguished by this method.

Author Response

Response to reviewers

Reviewer #2:

Comments and Suggestions for Authors

  1. The MS concerns distinguishing between three mosquito species - Culex pseudovishnui, Cx. tritaeniorhynchus, and Cx. vishnui - that belong to the Cx. vishnui species complex. However, the authors do not argue the need to distinguish between these species from an epidemiological point of view, as all three species are closely-related and all can spread Japanese encephalitis virus. Maybe they can add in the introduction what is the epidemic benefit of distinguishing between these species instead of identifying Cx. vishnui sensu lato?

- Many thanks, I added as you suggested in the “Introduction” section. “According to previous studies, species members of the Cx. vishnui subgroup exhibited distinct behaviors associated with disease outbreaks. For example, a survey of host selection or preference of mosquito species in Kandal province, Cambodia revealed that Cx. vishnui preferred humans more than animals, whereas Cx. tritaeniorhynchus preferred animals more than humans [13].”

  1. Overall, the study is done correctly, with the main result being that geometric morphometric (GM) analysis does not provide a guarantee of species differentiation in the complex. While I cannot evaluate morphological analysis, in my opinion a formal barcoding-gap analysis should be used for DNA-barcode analysis (https://bioinfo.mnhn.fr/abi/public/asap/, https://onlinelibrary. wiley.com/doi/10.1111/1755-0998.13281).

-Many thanks, I added the online assemble species by automatic partitioning (ASAP) approach for molecular species delimitation as you suggested. “For molecular species delimitation, the online assemble species by automatic partitioning (ASAP) approach based on the simple distance (p-distances) was useds (https://bioinfo.mnhn.fr/ abi/public/asap/).”

  1. Throughout the manuscript, the three species should be given in alphabetical order (Culex pseudovishnui, Cx. tritaeniorhynchus, and Cx. vishnui); the information that Cx. pseudovishnui and Cx. vishnui are more closely related is only the result of this study.

- Thank you very much. I fixed it as you suggested.

  1. L15, 64 – Culex should be abbreviated.

- Thank you very much. I fixed it as you suggested.

  1. The abstract (L36-37) is not consistent with the results and the conclusion that GM does not guarantee species differentiation.

- Thank you very much. I fixed it as you suggested. “Japanese encephalitis (JE) is a viral infection of the brain caused by the Japanese encephalitis virus, which spreads globally, particularly in 24 countries of Southeast Asia and the Western Pacific region. In Thailand, the primary vectors of JE are Cx. pseudovishnui, Cx. tritaeniorhynchus, and Cx. vishnui of the Cx. vishnui subgroup. The morphologies of three mosquito species are extremely similar, making identification challenging. Thus, geometric morphometrics (GM) and DNA barcoding were applied for species identification. The results of the GM based on wing shape revealed that this technique was not perfectly powerful for distinguishing Cx. pseudovishnui, Cx. tritaeniorhynchus, and Cx. vishnui (88.34% of total performance). On the contrary, the DNA barcoding yielded excellent results in identifying these Culex species (bootstrap support > 93%). However, in the absence of the required facilities for DNA barcoding, GM techniques can be employed in conjunction with morphological methods to enhance the reliability of species identification. Based on the results of this study, our approach can help guide efforts to identify members of the Cx. vishnui subgroup, which will be useful for the effective vector control of JE in Thailand.”

  1. L112, at the end of the sentence that COI is frequently used as the marker of choice for DNA barcoding you can add “including metabarcoding approach” (you can see and cite Trzebny et al. 2020, https://doi.org/10.1111/1755-0998.13205).

-Many thanks, I cited Trzebny et al. 2020, https://doi.org/10.1111/1755-0998.13205 as you suggested.

  1. L216-219 - It is better to remove gaps and ends (5’ and 3’) from the primer sequences, as they make sequences harder to copy for ordering primers.

- Thank you very much. I fixed it as you suggested.

  1. Line 218 – remove the word “mix”.

- Thank you very much. I fixed it as you suggested.

  1. L220 - Instead "0.2 μM forward primer, 0.2 μM reverse primer" just "0.2 μM each primer".

- Thank you very much. I fixed it as you suggested.

  1. L230 - remouve "nucleotide" just "sequencing".

- Thank you very much. I fixed it as you suggested.

  1. L235-237 - change to: "were compared to sequences published in GenBank (https://blast.ncbi.nlm.nih.gov/) and BOLD Systems (https://www.boldsystems.org/index.php)."

- Thank you very much. I fixed it as you suggested. “To confirm the identities our COI sequences, they were compared to sequences published in GenBank (https://blast.ncbi.nlm.nih.gov/) and BOLD Systems (https://www.boldsystems.org/index.php).”

  1. L248 – some fonts are bigger than others. 

- Thank you very much. I fixed it as you suggested.

  1. The caption to Figure 7 should be expanded to include information on what data was used and how they were analyzed.

- Thank you very much. I fixed it as you suggested. “Figure 7. Hierarchical clustering tree based on wing shape among Cx. pseudovishnui, Cx. tritaeniorhynchus and Cx. vishnui. The UPGMA algorithm is used to create a hierarchical clustering tree based on Mahalanobis distances between average group shapes.”

  1. L327 – sequences cannot be “approximately 735 bp” long. They have to be the same length (it is impossible that deletion/insertion of an amino acid can occur in such closely related species); they also should be trimmed to the same length in order not to introduce additional, non-existent sequence variability.

- Thank you very much. I fixed it as you suggested. “The length of the mitochondrial COI sequences analyzed was 735 bp.”

  1. L332-345 – Distances should include standard deviation (SD) value instead of a range. Also phylogenetic relationships could be better described (e.g. Cx. vishnui and Cx. pseudovishnui formed well supported clade which was sister to Cx. tritaeniorhynchus).

- Thank you very much. I fixed it as you suggested.

  1. Table 4 repeats content from L331-341.

- Thank you very much. I removed some of the information in the text so that it doesn't duplicate the table.

  1. Fig. 8 caption – remove “text of the” and leave only “The COI sequences...”. Why “Culex species” is underlined?

- Thank you very much. I fixed it as you suggested. “Neighbor-joining tree showing the phylogenetic relationships between Cx. pseudovishnui, Cx. tritaeniorhynchus and Cx. vishnui. The COI sequences generated in this study are black (n = 30), while those from GenBank are red (n = 6). The Cx. quinquefasciatus sequence from GenBank (OL743030) was used as an outgroup. Only bootstrap values ≥ 50% are shown. The result of species delimitation by the ASAP method is indicated by vertical red bars.”

  1. L390 – check the sentence; something is missing (“immature stage of 28].”). The next sentence is also unclear, and information is tautological.

- Thank you very much. I fixed it as you suggested. “The wing size is more sensitive to the environment, particularly the influence of temperature, humidity, and food availability in breeding sites during the development of the immature stage of mosquitoes [28]. In contrast, wing shape analyses yielded better outcomes than wing size.”

  1. L432 - and everywhere in the text where it fits - maybe it's a good idea to replace species identification with species distinguishing?

- Thank you very much. I fixed it as you suggested.

  1. The conclusions should be re-written. They should not provide results and should include a real finding that the GM is suitable for identifying Cx. vishnui sensu lato, but particular species in the complex cannot be unambiguously distinguished by this method.

- Thank you very much. I fixed it as you suggested. “The GM technique based on wing shape was not perfectly powerful for distinguishing Cx. pseudovishnui, Cx. tritaeniorhynchus, and Cx. vishnui (88.34% of total performance).”